# Trained recurrent neural networks develop phase-locked limit cycles in a working memory task

**Matthijs Pals** [1,2]*, **Jakob H. Macke** [1,2,3], **Omri Barak** [4,5]*

1 Machine Learning in Science, Excellence Cluster Machine Learning, University of Tübingen, Tübingen, Germany, 2 Tübingen AI Center, University of Tübingen, Tübingen, Germany, 3 Department Empirical Inference, Max Planck Institute for Intelligent Systems, Tübingen, Germany, 4 Rappaport Faculty of Medicine Technion, Israel Institute of Technology, Haifa, Israel, 5 Network Biology Research Laboratory, Israel Institute of Technology, Haifa, Israel

\* matthijs.pals@uni-tuebingen.de (MP); omri.barak@gmail.com (OB)

**Data Availability Statement:** We have made all code used to generate the results and figures available at: https://github.com/mackelab/phase-limit-cycle-RNNs.

## Abstract

Neural oscillations are ubiquitously observed in many brain areas. One proposed functional role of these oscillations is that they serve as an internal clock, or 'frame of reference'. Information can be encoded by the timing of neural activity relative to the *phase* of such oscillations. In line with this hypothesis, there have been multiple empirical observations of such *phase codes* in the brain. Here we ask: What kind of neural dynamics support phase coding of information with neural oscillations? We tackled this question by analyzing recurrent neural networks (RNNs) that were trained on a working memory task. The networks were given access to an external reference oscillation and tasked to produce an oscillation, such that the phase difference between the reference and output oscillation maintains the identity of transient stimuli. We found that networks converged to stable oscillatory dynamics. Reverse engineering these networks revealed that each phase-coded memory corresponds to a separate limit cycle attractor. We characterized how the stability of the attractor dynamics depends on both reference oscillation amplitude and frequency, properties that can be experimentally observed. To understand the connectivity structures that underlie these dynamics, we showed that trained networks can be described as two phase-coupled oscillators. Using this insight, we condensed our trained networks to a reduced model consisting of two functional modules: One that generates an oscillation and one that implements a coupling function between the internal oscillation and external reference.

In summary, by reverse engineering the dynamics and connectivity of trained RNNs, we propose a mechanism by which neural networks can harness reference oscillations for working memory. Specifically, we propose that a phase-coding network generates autonomous oscillations which it couples to an external reference oscillation in a multi-stable fashion.

**Funding:** This work was supported by the Deutsche Forschungsgemeinschaft (DFG, German Research Foundation) through SFB 1089 (Project-ID 227953431 to JHM) and Excellence Strategy (EXC number 2064/1 – 390727645 to JHM), the German Federal Ministry of Education and Research (BMBF) through the Tübingen AI Center (FKZ01IS18039A to JHM) and the ISRAEL SCIENCE FOUNDATION (grant No. 1442/21 to OB) and HFSP research grant (RGP0017/2021 to OB). The funders had no role in study design, data collection and analysis, decision to publish, or preparation of the manuscript.

**Competing interests:** The authors have declared that no competing interests exist.

## Author summary

Many of our actions are rhythmic—walking, breathing, digesting and more. It is not surprising that neural activity can have a strong oscillatory component. Indeed, such brain waves are common, and can even be measured using EEG from the scalp. Perhaps less obvious is the presence of such oscillations during non-rhythmic behavior—such as memory maintenance and other cognitive functions. Reports of these cognitive oscillations have accumulated over the years, and various theories were raised regarding their origin and utilization. In particular, oscillations have been proposed to serve as a clock signal that can be used for temporal-, or phase-coding of information in working memory. Here, we studied the dynamical systems underlying this kind of coding, by using trained artificial neural networks as hypothesis generators. We trained recurrent neural networks to perform a working memory task, while giving them access to a reference oscillation. We were then able to reverse engineer the learned dynamics of the networks. Our analysis revealed that phase-coded memories correspond to stable attractors in the dynamical landscape of the model. These attractors arose from the coupling of the external reference oscillation with oscillations generated internally by the network.

## Introduction

Rhythmic neural activity is an abundant phenomenon in nervous systems. Neural oscillations naturally underlie behavior with an observable oscillatory component, such as walking and digesting [1, 2]. Oscillating neural activity is also widely observed in brain regions implicated in higher cognitive functions, where there is no obvious correlate to oscillatory behavior [3]. Such rhythmic neural activity has been suggested to support short-term memory maintenance (among other functions), by serving as an internal clock for the brain [4–9]. This makes *phase coding* possible: Information can be encoded by spikes that are systematically timed with respect to the phase of ongoing oscillations. Empirical observations of a *phase code* in the brain were first described in the hippocampus of moving rats [10]. Since then, phase coding has also been associated with the representation of discrete object categories [11–13] for short-term maintenance of stimuli [14–18] and goals [19]. Such observations have been reported in a wide range of brain regions, including the human medial temporal lobe as well as primate prefrontal and sensory cortices.

Oscillations are a dynamic phenomenon, and it is therefore a natural question to ask: How do ongoing oscillations in the brain interact with the neural dynamics that support cognitive functions? Specifically, we seek to characterize dynamical systems that could underlie working memory relying on phase coding with neural oscillations. We do so by assuming that cognitive functions can be described by a low-dimensional dynamical system, implemented through populations of neurons (computation through dynamics) [20–26], in line with empirical observations [27, 28].

We make use of artificial recurrent neural networks (RNNs). RNNs are universal dynamical systems, in the sense that they can approximate finite time trajectories of any dynamical system [29, 30]. We are thus able to use these networks as hypothesis generators—first training them on a cognitive task, and then reverse engineering the resulting networks [17, 20, 24, 26, 31–38].

Concretely, we provide RNNs with both a reference oscillation, and stimuli. Experimental observations show that the phase of spikes relative to ongoing oscillations contains information about external stimuli [12, 14, 17–19]. Analogous to this, we ask the RNN to

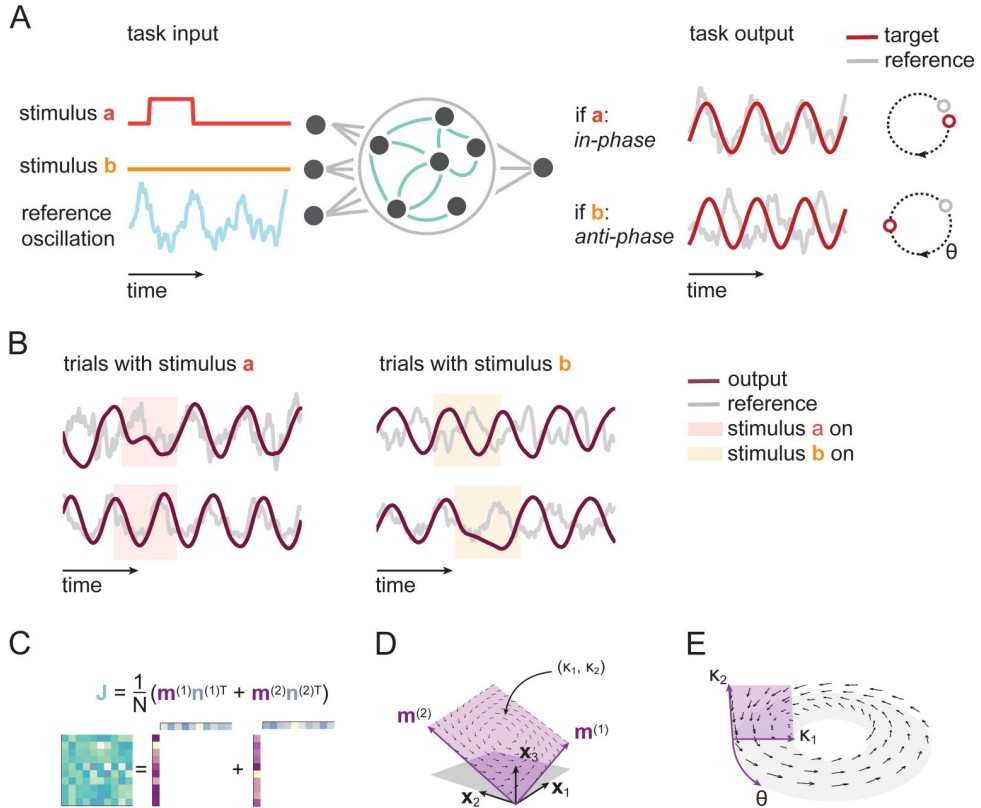

**Fig 1. Trained RNNs encode stimuli in oscillation phase.** A) RNNs receive transient stimuli as input, along with a reference oscillation. Networks are trained to produce an oscillation, such that the phase of the produced oscillation (relative to the reference oscillation), maintains the identity of transient stimuli. B) Example output of trained networks. Transient presentation of stimulus *a*, results in an in-phase output oscillation (left), regardless of the initial phase (top or bottom). Similarly, the *b* stimulus results in an anti-phase oscillation, again irrespective of its initial phase (right). C) To obtain a tractable model, we apply a low-rank constraint to the recurrent weight matrix of the RNN, i.e., we require that the weight matrix can be written as the outer product of two sets of vectors $\mathbf{m}^{(1)}$, $\mathbf{m}^{(2)}$ and $\mathbf{n}^{(1)}$, $\mathbf{n}^{(2)}$. D) Low-rank connectivity leads to low dimensional dynamics. In the absence of any input, the recurrent dynamics, described by coordinates $\kappa_1$, $\kappa_2$, lie in a linear subspace spanned by $\mathbf{m}^{(1)}$ and $\mathbf{m}^{(2)}$ (purple). E) When probing the model with sinusoidal oscillations, we can rewrite the system as a dynamical system in a three-dimensional phase space, where the additional axis, $\theta$, is the phase of the input oscillation. We can visualize this phase space as a toroid such that the horizontal circle represents $\theta$, and the vertical cross-section is the $\kappa_1$, $\kappa_2$ plane.

produce an oscillatory signal, which has to code the identity of stimuli by its phase relative to the reference oscillation (Fig 1). Depending on the training setup, we find two different solutions, of which one corresponds to network units coding for information by their phase and one solution in which units code with their average firing rates (Fig 2). We show that phase-coded memories correspond to stable limit cycles and demonstrate how empirically observable quantities control the stability of these attractors (Fig 3). Having detailed the dynamics, we study the connectivity of our trained RNNs (Fig 4). Finally, we show that the system can be well approximated by two coupled oscillators. Based on these analyses, we propose that phase-coded memories reside in stable limit cycles, resulting from the coupling of an oscillation generated by the recurrent network to an external reference oscillation.

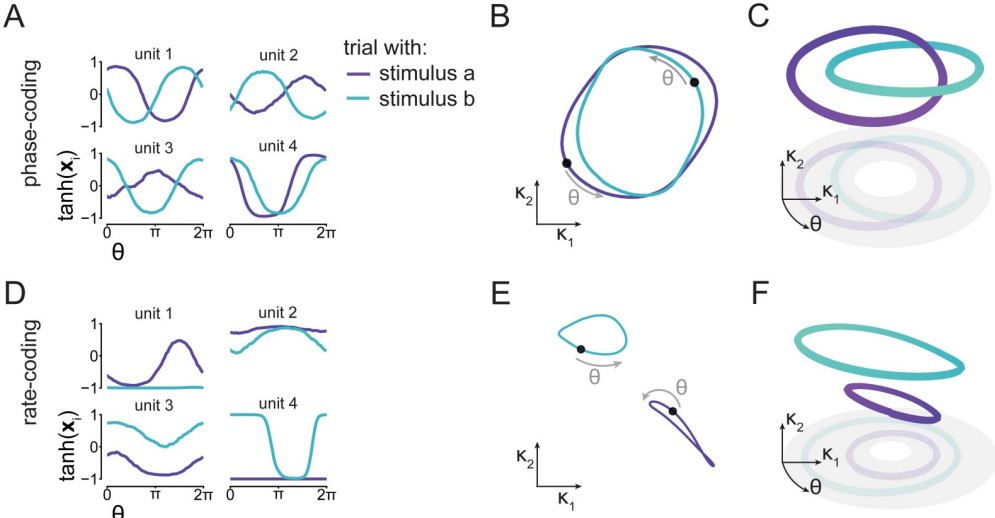

**Fig 2. Both phase coding and rate coding can solve the working memory task.** We found two qualitatively different solutions. A) In the first solution (panels A to C), we find that single unit activity codes for stimulus information by relative phase: We plot the rates $\tanh(x_i)$ of 4 example units $i$, as a function of the reference oscillation phase $\theta$. We find that single unit activity oscillates, with the phase relative to the reference oscillation depending on stimulus identity (colors). B) Projecting $\mathbf{x}$ to the $\kappa_1$, $\kappa_2$ plane reveals that population activity lies on overlapping cycles in this plane. Here, the black dots denote $\theta = 0$. C) In the full phase space $\mathcal{M}$, the trajectories are non-overlapping, but the cycles are linked. D) In the second solution (Panels D to F), we find that stimulus identity does not modulate the phase of single units, but rather their mean activity. E) This rate-code corresponds to two cycles separated in the $\kappa_1$, $\kappa_2$ plane. F) These cycles are also separated in the full phase space.

## Results

### Tractable, oscillating recurrent neural networks perform a working memory task

In order to study the dynamics of phase coding during working memory, we defined a task in which an RNN receives transient stimuli, and has to encode their identity using the relative phase of oscillations (Fig 1A). The network consists of $N$ units, with activation $\mathbf{x}(t) \in \mathcal{R}^N$, recurrently connected via a connectivity matrix $\mathbf{J} \in \mathcal{R}^{N \times N}$, and receiving external oscillatory input $u(t) \in \mathcal{R}$ [7, 18, 39], as well as stimuli $\mathbf{s}(t) \in \mathcal{R}^2$,

$$\tau \frac{d\mathbf{x}}{dt} = -\mathbf{x}(t) + \mathbf{J} \tanh(\mathbf{x}(t)) + \mathbf{I}^{(osc)} u(t) + \mathbf{I}^{(s)} \mathbf{s}(t) + \boldsymbol{\xi}(t), \tag{1}$$

where $\tau$ represents the time constant of the units, tanh is an elementwise non-linearity, $\mathbf{I}^{(osc)} \in \mathcal{R}^N$, $\mathbf{I}^{(s)} \in \mathcal{R}^{N \times 2}$ represent the input weights, and $\boldsymbol{\xi}(t) \in \mathcal{R}^N$ independent noise for each unit. Motivated by experiments observing phase coding relative to local field potential oscillations (LFPs) [12, 14, 15, 17–19], we used filtered rat CA1 local field potentials [40, 41] as input $u(t)$. By using the LFP as input, we make the assumption that the neurons in our model have an overall negligible effect on the LFP, which is valid if they are only a small subset of all contributing neurons, or because the LFP largely reflects input coming from an upstream population.

During a given trial, one of two stimuli $a$ or $b$, was transiently presented to the network at a randomized onset-time, with amplitude $s_a$ or $s_b$, respectively, representing the working-memory to be stored (e.g. a navigational goal [19], or sequence list position [17, 18]). To define the

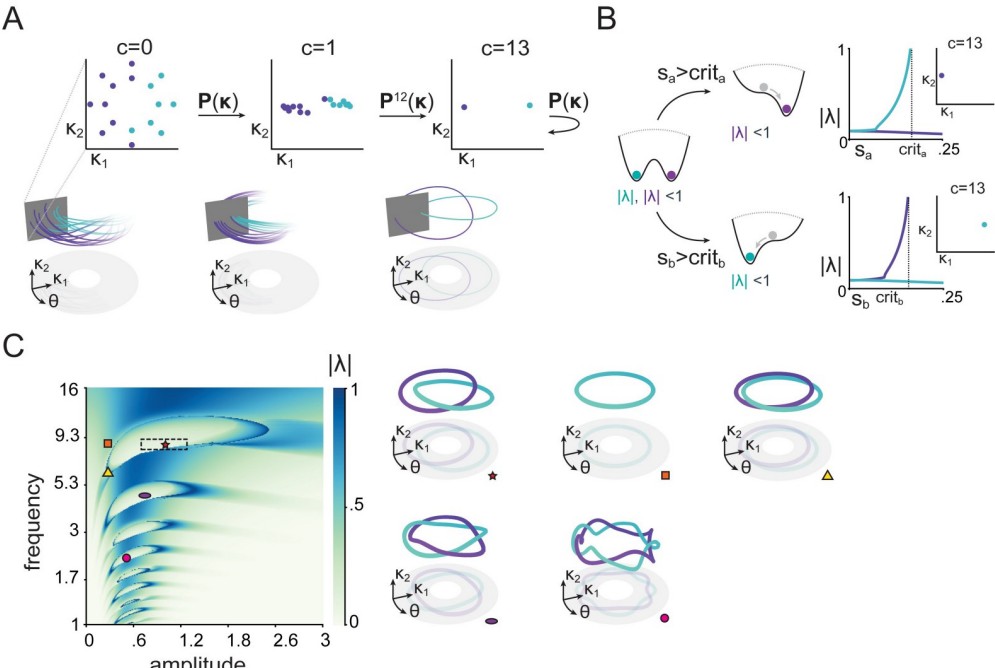

**Fig 3. Input controls the stability of the attractors in which phase-coded memories reside.** A) Poincaré maps illustrate that the cycles of Fig 2 correspond to attractive limit cycles. Sixteen trajectories with different initial conditions in three different snapshots (after 0, 1 and 13 cycles). Color marks the cycle they end up in. Bottom part shows how trajectories intersect the Poincaré plane, which is shown on top. B) Linear stability analysis of the Poincaré maps shows that stimuli of sufficient magnitude ($s_i > crit_i$) lead to a bifurcation, such that only one of two limit cycles remains stable. Left: cartoon of the bistable dynamics (without stimuli) and bifurcation during stimulus presentation illustrated as a potential well. Right: Floquet multiplier norm (a measure of stability) as a function of stimulus amplitude. Insets show the Poincaré map after 13 iterations with the stimulus presented at amplitude $s_i = 1$. C) Stability analysis for a range of amplitudes and frequencies of the reference oscillation (quantities that naturally vary in the brain). The box with dashed lines indicates the 5'th and 95'th percentiles of the amplitudes and frequencies used during training trials. The 'islands' correspond to regions where bistable dynamics persist. If the reference oscillation and amplitude are within such a bistable region (e.g. the red star), there are two stable cycles, and the model can maintain memory of a stimulus. For lower amplitudes (orange square and yellow triangle), the model only retains bistability if the frequency is also lower (yellow triangle). The additional 'islands' correspond to regions with bistable $m : n$ phase locking, where the reference oscillation is an integer divisor of the oscillation generated by the RNN (e.g. purple oval, bistable 1 : 2 phase locking; pink circle, bistable 1 : 4 phaselocking). Trajectories on the right correspond to the different markers in the parameter space on the left.

desired output, we were inspired by experimental studies showing that the relative timing of spikes to an ongoing oscillation is informative about sensory stimuli. Similarly to these observations, we wanted RNNs to encode information about external events relative to ongoing oscillations. To do this, the task for the network was to produce an oscillation, that is either in-phase with respect to the reference signal, following stimulus *a*, or anti-phase, following stimulus *b*. Networks were trained with backpropagation through time (S1 Fig) and successfully learned the task (Fig 1B).

As we were interested in reverse-engineering the networks, we made two simplifications. After training, we replaced the LFP reference signal with a pure sine wave with phase *θ*. During training, we chose a constraint on the connectivity that reduces the complexity of our analysis while still allowing for expressive networks. Specifically, we constrained the recurrent weight matrix to be of rank two, by decomposing it as an outer product of two pairs of vectors (Fig

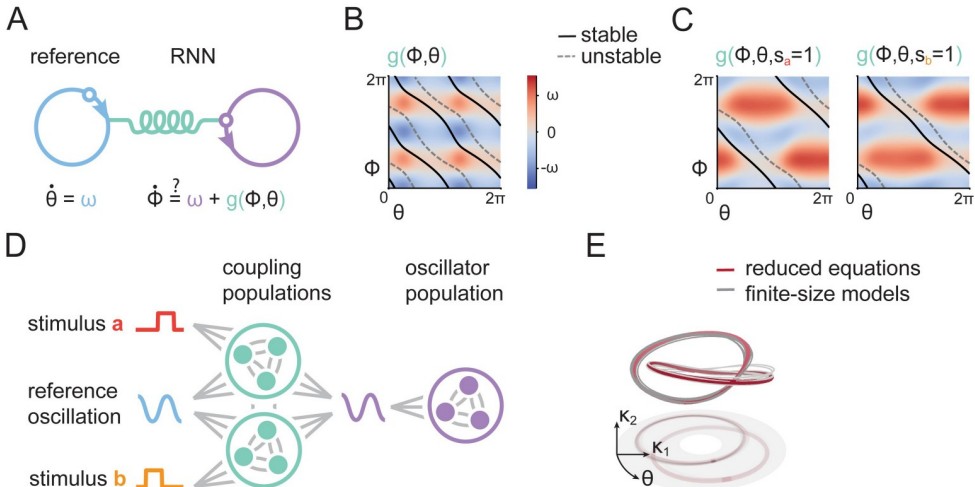

**Fig 4. Phase-coding RNNs are oscillators.** A) We hypothesized that the model functions as two coupled oscillators, where one represents the external reference oscillation phase and one the RNNs' internal oscillation phase. B) We extracted the coupling functions from our trained RNNs. This coupling function induces bistable dynamics when it couples two oscillators, as apparent from the superimposed stable and unstable trajectories. C) Input stimuli transiently modify the coupling function, resulting in the bifurcation, previously observed in Fig 3B. D) To formalize the coupled oscillator description, we created a reduced model where weights are drawn from a mixture of Gaussians. This model consists of a population that generates oscillations, and two populations that together implement the coupling function between internal and external oscillations. E) Simulating a reduced set of equations that describe the idealized dynamics of RNNs with connectivity in terms of a Gaussian mixture distribution, as well 10 finite-size models with weights sampled from this distribution, all result in trajectories similar to our original system, validating our reduced description.

1C; see S2 Fig for unconstrained networks) [26, 30, 42, 43],

$$\mathbf{J} = \frac{1}{N}\left(\mathbf{m}^{(1)}\mathbf{n}^{(1)\mathsf{T}} + \mathbf{m}^{(2)}\mathbf{n}^{(2)\mathsf{T}}\right). \tag{2}$$

By constraining the weight matrix, we directly constrain the dynamics. Specifically, the projections of network activity on the two vectors $\mathbf{m}^{(1)}$, $\mathbf{m}^{(2)}$, which we term $\kappa_1$, $\kappa_2$ respectively, are sufficient to describe the network dynamics in the absence of inputs (Fig 1D).

In the presence of sinusoidal input, the $\kappa$s are not sufficient to describe the dynamics, and we also need to know the current phase $\theta$ of the reference oscillation. These three numbers constitute the complete phase space $\mathcal{M}$ for our dynamical system (Fig 1E),

$$\mathcal{M} = \{(\kappa_1, \kappa_2, \theta) \in \mathcal{R}^2 \times \mathcal{S}^1\}.$$

## Phase-coded memories correspond to limit cycle attractors

We reverse-engineered the dynamics of our trained networks in order to understand how they solve the task [37]. We found that networks converge to one of two solutions, characterized by their activity following the transient stimuli. The solution that the network ends up choosing is dependent on both the training setup and the initialisation used (See S3 Fig for an investigation into when each solution arises).

In the first solution, individual network units code for stimuli by using their phase of oscillation relative to the reference, as illustrated by rate traces of example units (Fig 2A), as well as statistics for all units (S3 Fig). In $\kappa$ space, the population activity corresponds to two cycles that

roughly lie on the same area, but have a different phase relation to the reference oscillation (Fig 2B). In the full phase space $\mathcal{M}$ introduced above, this solution corresponds to the network activity residing in one of two linked cycles (Fig 2C).

In the second solution, single units code stimulus identity by their average activation (Fig 2D). Units with activity far away from 0 saturate their (i.e. are at the flat part of their) tanh nonlinearity, and as a consequence have little effect on the phase of the output oscillation. A different subset of units saturates after either stimulus, leading to different units determining the output oscillation. These units either directly transmit the input oscillation, creating an in-phase oscillation, or flip its sign to create an anti-phase oscillation. In $\kappa$ space, activity lies on two non-overlapping cycles (Fig 2E), which are likewise completely separated in the full phase space (Fig 2F).

Given the that the phase-coding solution matches the phenomenon we set out to investigate, whereas the rate-coding solution shows similarities to previous work on fixed point dynamics in RNNs [26, 30, 31], we focus here on the analysis of the 'phase-coding' solution (Fig 2A, 2B and 2C). The rate-coding and its similarities to existing work is detailed in the Supplementary (S4 Fig).

Single trials have a limited duration, and hence the cycles we observed might arise either from a transient dynamical phenomenon or from stable attractors. These would lead to different experimental observations of residual dynamics (trial-to-trial variability in neural population responses) [44], and responses to perturbations [36]. To study stability of limit cycles, we used discrete-time iterative maps, also known as Poincaré maps [45, 46]. To obtain such a map, we start by making a cross-section $Q$ of the phase space ($Q = \{(\kappa_1, \kappa_2, \theta) : \mathrm{mod}\ \theta = 2\pi\}$), such that trajectories go through $Q$ once every cycle of the reference oscillation (Fig 3A). We can then define an iterative map, corresponding to following a trajectory starting in $Q$ till its next intersection with $Q$. We can denote this mapping ($Q_0 \rightarrow Q_{2\pi}$) as

$$\boldsymbol{\kappa}_{c+1} = \mathbf{P}(\boldsymbol{\kappa}_c),$$

where $\boldsymbol{\kappa}_c = [\kappa_{1,c}, \kappa_{2,c}]^\mathsf{T}$ corresponds to the $c$'th intersection of trajectory with $Q$. Practically, we obtain the map by integrating the dynamics for one period of the input oscillation (Eq 9). Once we find a fixed point of $\mathbf{P}$, i.e. a trajectory comes back exactly where it started, we obtain a limit cycle of the original system.

Using the Poincaré map, we found that trajectories starting from many initial conditions quickly converged to one of two fixed points of $\mathbf{P}$ (Fig 3A), corresponding to the cycles observed during the working memory task (Fig 2C). To confirm their stability, we performed linear stability analysis by calculating the Floquet multipliers ($\lambda$), i.e., the eigenvalues of the linearized Poincaré map. Limit cycles are stable if these have a magnitude less than one, i.e., $|\lambda| < 1$.

This analysis allows us to study how the two inputs direct activity to the corresponding limit cycle. Without inputs, both cycles are stable (Fig 3B, leftmost cartoon). We can then toni-cally provide input corresponding to a scaled version of one of the stimuli, and recalculate the maximal Floquet multiplier for the resulting limit cycle. We found that this procedure gradu-ally destabilizes the other limit cycle, so that eventually only the limit cycle corresponding to the correct input remains (Fig 3B, right cartoons). Thus, for both stimuli, once a sufficient amplitude is crossed ($s_i > crit_i$; Fig 3B), a bifurcation occurs and only one limit cycle remains stable. When the transient stimulus is removed, both limit cycles are stable again, but the net-work has already been directed to one of them.

As realistic neural oscillations exhibit drifts in both frequency and amplitude, we next assessed the robustness of our network to variations of the reference oscillation, going

substantially beyond the range encountered during training. This analysis also makes testable predictions, as changes in oscillation amplitude and frequency can be experimentally observed, and potentially controlled [47].

We calculated the norm of the maximum Floquet multiplier for varying frequencies and amplitudes of the reference oscillation (Fig 3C), in order to determine when bifurcations occur (i.e., maximum multiplier norm exceeding one). The resulting diagram allows us to draw two conclusions: First, there are multiple regions with two stable oscillations, but with an $m : n$ phase coupling, i.e., where the internal oscillation frequency is an integer multiplier of the reference frequency. This kind of cross-frequency phase-phase coupling is an integral part of previously proposed theories of phase coding [7, 9] (however, interpretation of experimental observations can be challenging [48]). Second, based on the shape and location of the bistable regions (the 'islands' in Fig 3C), we predict that being able to phase-code stimuli is possible for reference oscillations with low frequency, only if the amplitude is also low.

## Low-rank RNNs as phase-coupled oscillators

Having described the dynamics, we next aimed to understand the population structure that gives rise to these dynamics. To proceed, we observe that, following the stimulus, the internal dynamics of the RNN quickly converge to a trajectory in the $(\kappa_1, \kappa_2)$ plane, which can be approximately described by its phase (Fig 2B $\phi = \arctan\left(\frac{\kappa_2}{\kappa_1}\right)$). Using this insight, we can rewrite our model as two phase-coupled oscillators (Fig 4A): one that describes the external reference oscillation phase ($\theta$) and one that characterizes the internal RNN phase ($\phi$). The dynamics of these oscillators are then determined by the oscillation frequency $\omega$, and a coupling function $g(\phi, \theta)$ [49]:

$$
\begin{aligned}
\dot{\theta} &= \omega, \\
\dot{\phi} &= \omega + g(\phi, \theta).
\end{aligned}
\tag{3}
$$

To extract the coupling function $g$ from our trained networks, we rewrote the model equations in polar coordinates, and approximated the radius in $\kappa$-space as constant (Fig 4B, Methods). We then simulated the oscillators of Eq 3 using this function, and observed two stable trajectories (superimposed lines in Fig 4B), showing that the coupling function is sufficient to induce bistable dynamics. We found that the convergent trajectories are close to those of the RNN when projected to the same phase space ($\theta, \phi$). We thus verified that our approximate description captures the dynamics of the full RNN (S5 Fig). Furthermore, the observed stimulus-induced dynamics (Fig 3C) can now be explained by the effect of input on the coupling function. In the presence of input, the coupling function only admits a single stable trajectory, resulting from one stable and one unstable trajectory colliding and annihilating in a saddle-node bifurcation (Fig 4C).

Finally, we wanted to know how the rank-two connectivity leads to these coupled oscillators. To this end, we approximated the weights in the connectivity vectors of the RNN using a mixture of Gaussians [26, 30, 42]. Fitting a mixture of Gaussians to the connectivity of trained networks [50], and sampling weights from this mixture, as in previous work [26], did not reliably lead to functioning networks (S6A Fig). We were, however, able to manually design a reduced model, with weights drawn from a mixture of Gaussians, based on the reverse-engineered dynamics of trained networks, as well as the structure in their connectivity as revealed through the clustering analysis (S6B Fig). The reduced model consisted of three mixture components (or subpopulations; Fig 4D and S7 Fig). One component is not connected to the reference oscillation and autonomously generates its own oscillation. The other two components

implement the required coupling function. The two coupling components differ only in their connectivity with the input; one population saturates (i.e. is inhibited) by stimulus *a*, and one by stimulus *b*. By adding additional coupling populations, one can obtain a model that stores more than two stimuli, each in a separate limit cycle (S8 Fig).

The description in terms of a mixture of Gaussians enables us to derive a reduced set of three equations, which describe the dynamics of the RNN, in the limit of infinite units (Reduced models). We confirmed that finite-size networks are appropriately described by this reduced description, by simulating trajectories of 10 RNNs (with N = 4096 units) sampled from the reduced connectivity (Fig 4E). Thus, we show that a sufficient connectivity for working memory through phase-coding entails two modules: an oscillator and a coupling function that induces bistable dynamics.

## Discussion

In this study, we raised a hypothesis about a potential dynamical mechanism underlying phase-coding with neural oscillations. Namely, that phase coding in recurrent networks can be implemented through multi-stable coupling of two oscillations, one internal and one external to the network. We arrived at the hypothesis by training RNNs on a working memory task, while supplying them with oscillatory reference input. The networks had to encode transient stimuli by producing an oscillation with a persistent phase shift with respect to the reference input.

Through reverse engineering the dynamics of trained RNNs, we found that phase-coded memories correspond to stable periodic attractors. These materialized in our models as linked cycles in phase space. The presence of attractive oscillatory dynamics, as opposed to marginally stable or transient trajectories, can be detected by analyzing residual dynamics from data [44] or through perturbation studies [36]. We showed how LFP frequency and amplitude jointly control the stability of the attractors. As LFP oscillation frequency and amplitude vary naturally in the brain, and can potentially be steered [47], this relationship could be probed directly from neural data.

Beyond characterizing the dynamics, we also revealed the effective underlying connectivity. We showed that our trained RNNs are analogous to coupled oscillators and that a sufficient connectivity for phase coding entails two modules: one generating an oscillation and one implementing a coupling function between this oscillation and an external reference one. Two independently generated oscillations that couple during memory are found in the medial temporal lobe (i.e. theta and gamma oscillations) [3, 5, 16]. Our model would predict that coupling functions extracted from neural data [49, 51], recorded from subjects performing working memory tasks, should have a structure that induces multi-stability.

We used trained recurrent neural networks [20, 22, 24]. RNNs can be trained to reproduce both neural activity and behavior [17, 23, 31, 34, 36]. The resulting networks can be understood in terms of their dynamical objects, i.e. the set of attractors [37, 52], and in terms of the trajectories [35, 53] flowing between them. In particular, recurrent models that implement discrete memory, as in this study, often have a separate *static* attractor, or fixed point, for each memory [30, 32, 38, 54], although oscillatory models of memory have also been proposed [17, 55–57]. Here we complement previous work by showing that stable *dynamic* attractors naturally emerge when tasking networks to store memories in the relative phase of oscillations.

Our findings support the notion that rhythmic neural activity can play a supporting role in cognitive phenomena. Brain waves during working memory are widely observed in neural systems of rats, primates, and birds [58]. Phase coding of information relative to theta oscillations has been proposed to be a general coding scheme in the brain [5]. Here, we focused on the

coding of two distinct stimuli in phase, which suffices to highlight the dynamics underlying the coding of discrete pieces of information. Both our model and findings can also be extended to coding for more than two items. We show that a rank-2 model with additional coupling sub-populations can memorise 4 stimuli in S8 Fig. Phase coding has also been observed during memory of sequences of information [17], as well as for coding of position [10]. Linking these findings to our model requires further investigation. We note that phase precession, in the sense of a continuous variable (e.g., position in a place field) changing the phase to which a group of neurons locks could be explained by our model through input translating the coupling function, as is tentatively shown in S9 Fig.

In summary, we used RNNs as trainable dynamical systems to form a hypothesis on how oscillations can be harnessed for neural computation. We proposed that phase-coded memories in recurrent networks reside in stable limit cycles resulting from the coupling of internal and external oscillations.

## Models and methods

### Training RNNs on a phase-coding task

**Model definition.**   As stated in Eq 1, we used a continuous time RNN with $N$ units [33],

$$\tau \frac{d\mathbf{x}}{dt} = -\mathbf{x}(t) + \mathbf{J}\tanh(\mathbf{x}(t)) + \mathbf{I}^{(osc)}u(t) + \mathbf{I}^{(s_b)}s_b(t) + \mathbf{I}^{(s_a)}s_a(t) + \boldsymbol{\xi}(t). \tag{4}$$

The input weights $\mathbf{I}$ of Eq 1 are split into three vectors in $\mathcal{R}^N$: The oscillatory reference input $u(t)$ with input weights $\mathbf{I}^{(osc)}$ and transient stimuli $s_a(t)$ and $s_b(t)$ with weights $\mathbf{I}^{(s_a)}$ and $\mathbf{I}^{(s_b)}$, respectively. To obtain networks with tractable dynamics, we implemented a low-rank constraint according to Eq 2. Trained networks consisted of $N = 512$ units, with $\tau = 20$ ms.

**Task.**   We defined a task in which stimulus identity is to be encoded in the phase of an oscillation. During each trial, of duration $T = 0.8s$, a reference oscillation $u(t)$ with random initial phase $\theta$ was provided to the RNN. Trials started with an initial period, with duration drawn uniformly from $[0.125s, 0.25s]$. After the initial period, either stimulus $a$, $s_a(t)$, or stimulus $b$, $s_b(t)$, was shown, consisting of a 'pulse' with a constant amplitude of 1 (see Fig 1A). The duration of stimuli was drawn uniformly from $[0.125s, 0.175s]$. For the remainder of the trial, the target $\hat{r}(t)$ was either $\sin(\theta(t) - 0.2\pi)$, when stimulus $a$ was shown ('in-phase' trial), or else $\sin(\theta(t) - 1.2\pi)$ when stimulus $b$ was shown ('anti-phase' trial).

**LFP processing.**   We used a publicly available data set containing LFP recordings from 3 Long-Evans rats, to obtain a reference oscillation that captures the statistics of ongoing oscillations in biological neural systems [40, 41]. Rats were chasing randomly placed drops of water or food, and neural activity was recorded using multichannel silicon probes inserted in area CA1 of the right dorsal hippocampus. The data contains LFP from 31 to 64 channels, recorded over multiple sessions.

We read the data using Neo [59]. We first re-sampled the data from 1250 Hz to 500 Hz and then high-pass filtered at 7 Hz using a FIR filter with Hamming window and 511 taps [60]. We normalised the signal by dividing the signal channel-wise with $\sqrt{2}\sigma_{LFP}$, where $\sigma_{LFP}$ is the channel-wise standard deviation, resulting in the signals having a root mean square equal to a sine wave with amplitude 1.

In order to obtain a unique reference signal for each training trial, we split the recordings for each rat in chunks. In particular, we first split the data into 4 second segments, and picked a random channel for each segment. To extract instantaneous phase of the LFP oscillation for creating training targets, we convolved the signal with complex Morelet wavelets, consisting of the product of a complex sinusoid with a Gaussian with standard deviation $c/2\pi f$. We picked

frequencies *f* from 7 to 9 in steps of 0.2, and set *c* ('cycles') to 7. For each trial we took the phase (angle) corresponding to the frequency with the highest power. The first second of the signal was discarded to avoid boundary effects. We also discarded a small fraction of trials with artifacts, i.e. trials where the maximum absolute value of the signal was larger than 4 (after normalisation). We ended up with 5708 trials from rat 1, 5669 from rat 2, and 5632 from rat 3.

**Training.** We approximated Eq 4 using the Euler–Maruyama method with timestep *h*, giving us for a rank-two network,

$$
\begin{aligned}
\mathbf{x}_{n+1} =\ & (1 - \frac{h}{\tau})\mathbf{x}_n \\
& + \frac{h}{\tau}\Big(\frac{1}{N}\sum_{i=1}^{2}\big[\mathbf{m}^{(i)}(\mathbf{n}^{(i)\mathsf{T}}\tanh(\mathbf{x}_n))\big] + \mathbf{I}^{(osc)}u_n + \mathbf{I}^{(s_b)}s_{b,n} + \mathbf{I}^{(s_a)}s_{a,n}\Big) \\
& + \sqrt{\frac{h}{\tau}}\mathcal{N}(0, 2\sigma_{noise}^2).
\end{aligned}
$$

We use *h* = 2 for training, whereas for the stability analysis and Figures, we take *h* = 0.5. We defined a mean-squared error loss $\mathcal{L}$ between the targets $\hat{r}(t)$ and a linear readout $r(t)$ of the model's activity at time *t*,

$$
\begin{aligned}
r(t) \ &= \frac{1}{N}\mathbf{w}^{\mathsf{T}}\mathbf{x}(t), \\
\mathcal{L} \ &= \frac{1}{T - T_s}\int_{T_s}^{T}(r(t) - \hat{r}(t))^2 dt,
\end{aligned}
\tag{5}
$$

where $T_s$ is the time of the stimulus offset and *T* is the time at the end of a trial. Note that, although we only demand that a weighted sum of the *activity* $\mathbf{x}(t)$ codes for the information, this also causes the *rates* $\tanh(\mathbf{x}(t))$ to phase-code for stimuli (Fig 2A). In principle, one could also directly read-out the rates during training: $r(t) = \frac{1}{N}\mathbf{w}^{\mathsf{T}}\tanh(\mathbf{x}(t))$. Depending on the initialisation and training setup, this can either lead to a rate-coding solution (Fig 2D, 2E and 2F), or a phase-coding solution (see S3 Fig for a more in-depth investigation).

We drew initial entries in the connectivity vectors from a zero-mean normal distribution, with a covariance matrix that is the identity matrix, except for an initial covariance between $\mathbf{m}^{(1)}$, $\mathbf{n}^{(1)}$ of .6, an initial covariance between $\mathbf{m}^{(2)}$, $\mathbf{n}^{(2)}$ of 0.6 and a variance for $\mathbf{w}$ of 16 [26]. Note that the initial covariances also have an effect on the solution the model finds (S3 Fig). We minimised Eq 5, by optimising with stochastic gradient descent all entries in $\mathbf{I}^{(osc)}$, $\mathbf{I}^{(s_a)}$, $\mathbf{I}^{(s_b)}$, $\mathbf{m}^{(1)}$, $\mathbf{m}^{(2)}$, $\mathbf{n}^{(1)}$, $\mathbf{n}^{(2)}$, as well as a scalar multiplying the readout weights $\mathbf{w}$. We used the Adam optimizer in Pytorch, with default decay rates (0.9, 0.999) and learning rate 0.01 [61, 62].

Each RNN was trained with LFP data from one rat. Shown in the main text is an RNN trained using LFPs from rat 2, whereas for the analysis of many trained models (S3 Fig), also a third of the RNNs were trained using data from rat 1, and a third on data from rat 3. Before each training run, we split the LFP into short segments, of which 90% were used as reference signal for the RNN during training trials, and 10% as reference signal during validation trials. The validation and training trials only differ in the portion of the local field potential that was used as reference signal for the network, and in the seed used for generating the randomised stimulus onsets and offsets. For the models shown in the main text, we trained for 50 epochs, in batch sizes of 128, where one epoch denotes all training trials created from the LFP of one rat. Due to the simplicity of the task, all networks we used in the main Figures converged at this point (S1 Fig), and we did not do any model selection.

Some of the models in the supplementary Figures (S2A, S3 and S8A Figs), were guided to a phase-coding solution by adding a regularisation term Reg to the loss, which keeps the average firing rates close to 0:

$$\text{Reg} = \frac{1}{N}\sum_i^N \left(\frac{1}{T-T_s}\int_{T_s}^T \mathbf{x}_i(t)dt\right)^2.$$ (6)

## Analysing dynamics of oscillating low-rank RNNs

**Dimensionality and dynamics of low-rank RNNs with periodic input.** Here, we show that the dynamics of a rank-2 RNN with periodic input lie on a 3 dimensional manifold. To facilitate the analysis, we first consider the dynamics of the network in the absence of transient stimuli ($s_a(t) = s_b(t) = 0$), and no recurrent noise ($\sigma_{noise} = 0$). Furthermore we set $\mathbf{m}^{(1)} \perp \mathbf{m}^{(2)}$, which one can always obtain by singular value decomposition of $\mathbf{J}$, even if during training $\mathbf{m}^{(1)}$ and $\mathbf{m}^{(2)}$ became correlated. We split up $\mathbf{I}^{(osc)}$ in the parts parallel and orthogonal to the $\mathbf{m}$'s,

$$\mathbf{I}^{(osc)} = \mathbf{I}_\perp + \mathbf{m}^{(1)}\alpha_1 + \mathbf{m}^{(2)}\alpha_2.$$

We can then express $\mathbf{x}(t)$ in the orthogonal basis

$$\mathbf{x}(t) = \mathbf{m}^{(1)}\kappa_1(t) + \mathbf{m}^{(2)}\kappa_2(t) + \mathbf{I}_\perp v(t),$$

with $i \in \{1, 2\}$:

$$\kappa_i(t) = \frac{\mathbf{m}^{(i)\mathsf{T}}\mathbf{x}(t)}{\mathbf{m}^{(i)\mathsf{T}}\mathbf{m}^{(i)}},$$

$$v(t) = \frac{\mathbf{I}_\perp^\mathsf{T}\mathbf{x}(t)}{\mathbf{I}_\perp^\mathsf{T}\mathbf{I}_\perp}.$$

Here $v(t)$ is the reference oscillation filtered by the model's time constant $\tau$. Using $*$ to denote convolution,

$$v(t) = \frac{1}{\tau}u(t) * e^{-\frac{t}{\tau}} + v(0)e^{-\frac{t}{\tau}}.$$

As both $u(t)$ and $v(t)$ explicitly depend on time, we first obtain the non-autonomous dynamical system $\mathbf{F}$,

$$\frac{d\kappa_1}{dt}, \frac{d\kappa_2}{dt} = \mathbf{F}(t, \kappa_1, \kappa_2),$$ (7)

with

$$\tau\frac{d\kappa_i}{dt} = -\kappa_i(t) + \frac{1}{N}\mathbf{n}^{(i)\mathsf{T}}\tanh(\mathbf{x}(t)) + \alpha_i u(t).$$ (8)

Next, we show that for periodic input this system can be considered as a three dimensional autonomous dynamical system. We introduce the variable $\theta = wt \bmod 2\pi$ such that

$$\frac{d\theta}{dt} = \omega.$$

We take $u(t) = \sin(\theta)$ and we can now find the closed form solution for $v$,

$$v(t) = \frac{1}{\sqrt{(w\tau)^2 + 1}} \sin(\theta - \arctan(w\tau)) + ce^{-\frac{t}{\tau}}.$$

Here, the last term on the right hand side will decay to 0. Practically, we get rid of this depen-dence on $t$, by taking appropriate $v(0)$ such that $c = 0$ (or by assuming the simulation has run for a little amount of time).

Following this definition, both $u$ and $v$ are functions of $\theta$, and we can consider the autono-mous dynamical system with coordinates $\kappa_1, \kappa_2, \theta$,

$$\frac{d\kappa_1}{dt}, \frac{d\kappa_2}{dt}, \frac{d\theta}{dt} = \mathbf{G}(\kappa_1, \kappa_2, \theta).$$

**Coordinate system for phase space figures.** To create the phase space figures, we applied the following coordinate transform, where $\tilde{x}, \tilde{y}, \tilde{z}$ denote the coordinates in the figure,

$$\tilde{x} = \cos(\theta)(\tilde{r} - \kappa_1),$$
$$\tilde{y} = \sin(\theta)(\tilde{r} - \kappa_1),$$
$$\tilde{z} = \kappa_2.$$

Given an initial condition for $\kappa_1$, i.e. $\kappa_1(0)$, one can always pick an $\tilde{r}$ such that this is an injec-tive map for all $t$. This requires $\tilde{r} < \kappa_1(t)$ for all $t$ (otherwise trajectories would cross at the cen-ter). From Eq 8, we can see that $\kappa_1(t) > \frac{1}{N}|\mathbf{n}^{(1)}|_1 + |\alpha_1| \Rightarrow \frac{d\kappa_1}{dt} < 0$, thus if we pick $\tilde{r} > \kappa_1(0) \geq \frac{1}{N}|\mathbf{n}^{(1)}|_1 + |\alpha_1|$, the requirement is satisfied.

**Poincaré maps and linear stability analysis.** We used Poincaré maps to study the stabil-ity of limit cycles [45, 46]. We took a cross section $Q = \{(\kappa_1, \kappa_2, \theta) : \mod \theta = 2\pi\}$, and cre-ated the iterative map from $Q$ to itself ($Q_0 \rightarrow Q_{2\pi}$),

$$\boldsymbol{\kappa}_{c+1} = \mathbf{P}(\boldsymbol{\kappa}_c),$$

where $\boldsymbol{\kappa}_c = \kappa_{1,c}, \kappa_{2,c}$ corresponds to the $c$'th intersection. Limit cycles correspond to fixed points $\boldsymbol{\kappa}^*$ for which

$$\boldsymbol{\kappa}^* = \mathbf{P}(\boldsymbol{\kappa}^*).$$

To study the stability of limit cycles we see what happens to a small perturbation $\eta_0$ to $\boldsymbol{\kappa}^*$. Applying a Taylor expansion around $\boldsymbol{\kappa}^*$, we have after one cycle,

$$\boldsymbol{\kappa}^* + \eta_1 = P(\boldsymbol{\kappa}^*) + D_{\boldsymbol{\kappa}}\mathbf{P}(\boldsymbol{\kappa}^*)\eta_0 + h.o.t.,$$

where $D_{\boldsymbol{\kappa}}$ denotes the gradient operator, the partial derivatives with respect to $\boldsymbol{\kappa}$. $D_{\boldsymbol{\kappa}}\mathbf{P}(\boldsymbol{\kappa}^*)$ is called the linearised Poincaré map at $\mathbf{k}^*$. To first order, perturbations scale proportional to the norm of its eigenvalues $\lambda$, called the Floquet (or characteristic) multipliers. To see this we can express $\eta_0$ as a linear combination of eigenvectors $\mathbf{e}$ of $D_{\boldsymbol{\kappa}}\mathbf{P}(\boldsymbol{\kappa}^*)$ (for some scalars $v$) and get the following expression for the perturbation after $c$ cycles,

$$\eta_c = \sum_{i=1}^{2} v_i \mathbf{e}_i \lambda_i^c.$$

We now show how to obtain $D_{\boldsymbol{\kappa}}\mathbf{P}(\boldsymbol{\kappa}^*)$. Let $\psi(t, \boldsymbol{\kappa_0})$ denote the flow, a mapping from $(t, \boldsymbol{\kappa})$ to $\boldsymbol{\kappa}$, obtained by integrating $\mathbf{F}(t, \boldsymbol{\kappa})$ (Eq 7) for duration $t$, with initial conditions $\boldsymbol{\kappa}_0$. The Poincaré

map then corresponds to [63]

$$
\begin{aligned}
\mathbf{P}(\boldsymbol{\kappa}_c) &= \psi(\mathcal{T}, \boldsymbol{\kappa}_c) \\
&= \int_0^{\mathcal{T}} \mathbf{F}(0, \psi(t, \boldsymbol{\kappa}_c))dt + \psi(0, \boldsymbol{\kappa}_c),
\end{aligned}
\tag{9}
$$

where the second line is obtained by applying the fundamental theorem of calculus. $\mathcal{T}$ denotes one period of the reference oscillation. Defining $\mathbf{M}(t)$ as $D_{\boldsymbol{\kappa}}\psi(t, \boldsymbol{\kappa}_c)$, one can obtain a variational equation for the linearized Poincaré map,

$$
\begin{aligned}
D_{\boldsymbol{\kappa}}\mathbf{P}(\boldsymbol{\kappa}_c) &= \int_0^{\mathcal{T}} D_{\psi}\mathbf{F}(0, \psi(t, \boldsymbol{\kappa}_c))D_{\boldsymbol{\kappa}_c}\psi(t, \boldsymbol{\kappa}_c)dt + \mathbf{I} \\
&= \mathbf{M}(\mathcal{T})
\end{aligned}
$$

with

$$
\frac{d\mathbf{M}(t)}{dt} = D_{\psi}\mathbf{F}(t, \psi(t, \boldsymbol{\kappa}_c))\mathbf{M}(t).
$$

Here, $\mathbf{M}(0) = \mathbf{I}$. $\mathbf{M}(t)$ is called the circuit or monodromy matrix. Since we can not obtain the circuit matrix analytically we approximated $\mathbf{M}(\mathcal{T})$ using the Euler method with timestep $h$.

## Coupled oscillators and population structure

**Extracting the coupling function.**   We can rewrite the internal dynamics of the RNN in polar coordinates with

$$
\kappa_1 = r\cos(\phi), \kappa_2 = r\sin(\phi).
$$

We extracted the coupling function $g$,

$$
\begin{aligned}
\tau\frac{d\phi}{dt} &= \frac{1}{r^2}\left(\frac{1}{N}(\kappa_1\mathbf{n}^{(2)} - \kappa_2\mathbf{n}^{(1)})^{\mathsf{T}}\tanh(\mathbf{x}) + (\kappa_1\alpha_2 - \kappa_2\alpha_1)u(\theta)\right), \\
g(\theta, \phi) &= \frac{d\phi}{dt} - \omega.
\end{aligned}
$$

Note that the radius of the two stable limit cycles might not be exactly constant, or equal to each other, so for Fig 3B and 3C we took for $r$ the mean radius over the two cycles.

**Reduced models.**   In order to find a simplified description of the dynamics of our models we assumed entries in the weight vectors of our network are $N$ samples, indexed by $j$, from a probability distribution $p(\mathbf{y})$ [26, 30, 42, 43]. To keep the equations brief, we assume one input $v$ here, which can be straightforwardly extended to multiple inputs. Then we have

$$
\begin{aligned}
\mathbf{I}_{\perp j}, \mathbf{n}_j^{(1)}, \mathbf{n}_j^{(2)}, \mathbf{m}_j^{(1)}, \mathbf{m}_j^{(2)} &\sim p(\mathbf{y}), \\
p(\mathbf{y}) &= p(I, n^{(1)}, n^{(2)}, m^{(1)}, m^{(2)}).
\end{aligned}
$$

We can then see Eq 8 as a sampling estimator for the recurrent input, which as $N \to \infty$ approaches the expectation,

$$
\tau\frac{d\kappa_i}{dt} \overset{N\to\infty}{=} -\kappa_i + \mathbb{E}_{p(\mathbf{y})}[n^{(i)}\tanh(\kappa_1 m^{(1)} + \kappa_2 m^{(2)} + v_1 I)].
$$

We assume $p(\mathbf{y})$ is a mixture of $L$ zero-mean Gaussians [26],

$$p(\mathbf{y}) = \sum_{l}^{L} w_l \mathcal{N}(0, \Sigma_l).$$

We can then obtain a mean-field description for the dynamics (see previous studies [26, 30, 42, 43] for a derivation). Here the effective connectivity consists of the covariances of the mixture components, modulated by a nonlinear 'gain' function (which approaches 0 when the non-linearity saturates),

$$\tau \frac{d\kappa_i}{dt} = -\kappa_i + \sum_{l}^{L} w_l \big(\kappa_1 \sigma^{(l)}_{m^{(1)}n^{(i)}} + \kappa_2 \sigma^{(l)}_{m^{(2)}n^{(i)}} + \nu \sigma^{(l)}_{In^{(i)}}\big) \mathbb{E}_{p(z)}\big[\tanh'(\sqrt{\Delta^{(l)}}z)\big],$$

with $\Delta^{(l)} = \big(\sigma^{(l)}_{m^{(1)}}\kappa_1\big)^2 + \big(\sigma^{(l)}_{m^{(2)}}\kappa_2\big)^2 + \big(\sigma^{(l)}_I \nu\big)^2$ and $z = \mathcal{N}(0,1)$.

Instead of numerically approximating the expectation as in previous studies [26, 30], we substitute $\mathrm{erf}\big(\frac{\sqrt{\pi}}{2}x\big)$ for $\tanh(x)$ and analytically obtain a simpler approximation.

$$
\begin{aligned}
\mathbb{E}_{p(z)}\big[\tanh'(\sqrt{\Delta}z)\big] \quad &\approx \mathbb{E}_{p(z)}\big[\mathrm{erf}'(\frac{\sqrt{\pi\Delta}}{2}z)\big] \\
&\approx \mathbb{E}_{p(z)}\big[e^{-\frac{\pi\Delta}{4}z^2}\big] \\
&\approx \frac{1}{\sqrt{1 + \frac{\pi}{2}\Delta}}.
\end{aligned}
$$

This gives us the following mean-field description for the rank-2 RNNs,

$$
\begin{aligned}
\tau \frac{d\kappa_i}{dt} \quad &= -\kappa_i + \sum_{l}^{L} w_l \big(\kappa_1 \sigma^{(l)}_{m^{(1)}n^{(i)}} + \kappa_2 \sigma^{(l)}_{m^{(2)}n^{(i)}} + \nu \sigma^{(l)}_{In^{(i)}}\big) \frac{1}{\sqrt{1 + \frac{\pi}{2}\Delta^{(l)}}}, \\
\Delta^{(l)} \quad &= \big(\sigma^{(l)}_{m^{(1)}}\kappa_1\big)^2 + \big(\sigma^{(l)}_{m^{(2)}}\kappa_2\big)^2 + \big(\sigma^{(l)}_I \nu\big)^2.
\end{aligned}
\tag{10}
$$

**Connectivity for mean-field model.**   By writing Eq 10 in polar coordinates we can see that the total change in phase of the model is the sum of the change in phase of its populations,

$$\tau \frac{d\phi(t)}{dt} = \sum_{l}^{L} w_l \frac{d\phi^{(l)}(t)}{dt}.
\tag{11}$$

Based on reverse engineering the dynamics and connectivity of our trained networks, we can now choose covariances in order to design a model that approximates the coupled oscillator equations (Eq 3). We here assume for all populations $l$, $\sigma^{(l)}_{m^{(2)}} = \sigma^{(l)}_{m^{(1)}}$ (denoted as $\sigma^{(l)}_m$).

We first initialize a population that generates oscillations with angular velocity $\omega$. We set this population unconnected to the input ($\sigma^{(i)}_I = 0$ for all $i$). When taking $\sigma_{m^{(2)}n^{(1)}} = -\sigma_{m^{(1)}n^{(2)}}$ and $\sigma_{m^{(1)}n^{(1)}} = \sigma_{m^{(2)}n^{(2)}}$ [30, 42, 55] this populations produces oscillations, with constant

frequency,

$$\frac{d\phi^{(p1)}(t)}{dt} = \sigma_{n^{(1)}m^{(2)}}^{(p_1)} \frac{1}{\sqrt{1 + \frac{\pi}{2}\Delta^{(p_1)}}},$$

$$\Delta^{(p1)} = \sigma_m^{2(p_1)}r^2.$$

Next, we create two populations that together implement the coupling function,

$$g(\phi, \theta) \propto \cos(2\phi)\frac{1}{\sqrt{1 + \frac{\pi}{2}\Delta}},$$

$$\Delta = \sigma_m^2 r^2 + \sigma_{I^{(osc)}}^2 \nu(\theta)^2. \tag{12}$$

Based on Fig 3B, it seems reasonable to assume that the bistable dynamics stem from $\sin(2\phi)$, $\cos(2\phi)$ and $\sin(2\theta)$, $\cos(2\theta)$ terms, which we obtained by setting $\sigma_{n^{(1)}m^{(2)}} = \sigma_{n^{(2)}m^{(1)}}$ (or $\sigma_{n^{(1)}m^{(1)}} = -\sigma_{n^{(2)}m^{(2)}}$), and $\sigma_{I^{(osc)}} \neq 0$.

In order to get the stimulus-induced bifurcation observed in Figs 2C and 3C, we, additionally set the following connectivity for the input to the coupling populations, $\sigma_{I^{(s_a)}}^{(p_2)} \neq 0, \sigma_{I^{(s_b)}}^{(p_3)} \neq 0, \sigma_{I^{(osc)}n^{(1)}}^{(p_2)} = -\sigma_{I^{(osc)}n^{(2)}}^{(p_2)} = -\sigma_{I^{(osc)}n^{(1)}}^{(p_3)} = \sigma_{I^{(osc)}n^{(2)}}^{(p_3)}$. Then the equations for the coupling populations read,

$$\frac{d\phi^{(p2)}(t)}{dt} = \left(\frac{\sqrt{2}}{r}\sigma_{n^{(1)}I^{(osc)}}^{(p_2)}\sin(\phi + \frac{1}{4})\nu(\theta) + \sigma_{n^{(1)}m^{(2)}}^{(p_2)}\cos(2\phi)\right)\frac{1}{\sqrt{1 + \frac{\pi}{2}\Delta^{(p_2)}}},$$

$$\Delta^{(p_2)} = \sigma_m^{2(p_2)}r^2 + \sigma_{I^{(osc)}}^{2(p_2)}\nu(\theta)^2 + \sigma_{I^{(s_a)}}^{2(p_2)}s_a(t)^2,$$

$$\frac{d\phi^{(p3)}(t)}{dt} = \left(\frac{\sqrt{2}}{r}\sigma_{n^{(2)}I^{(osc)}}^{(p_3)}\sin(\phi - \frac{3}{4})\nu(\theta) + \sigma_{n^{(1)}m^{(2)}}^{(p_3)}\cos(2\phi)\right)\frac{1}{\sqrt{1 + \frac{\pi}{2}\Delta^{(p_3)}}},$$

$$\Delta^{(p_3)} = \sigma_m^{2(p_3)}r^2 + \sigma_{I^{(osc)}}^{2(p_3)}\nu(\theta)^2 + \sigma_{I^{(s_b)}}^{2(p_3)}s_b(t)^2.$$

When both $s_a$ and $s_b$ are zero, the $\sin(\phi)$ terms cancel out and we retrieve the desired coupling function (Eq 12; where the $\cos(2\phi)$ terms dominate). When either stimulus $a$ or $b$ is on, population 2 or 3 is inhibited, respectively (the non-linearities of the units in this population saturate), causing the term on the right to go to zero. Then the $\sin(\phi)$ term of the population that is unaffected by the stimulus takes over the coupling function, mimicking what we saw in Fig 3B and 3C. For exact values of the parameters, see S7 Fig.

## Supporting information

**S1 Fig. Loss curves.** Loss over epochs for three models, each trained with LFP data from a separate rat. An epoch denotes one pass through all trials in the training or validation set. The validation trials are defined before training and are not used for calculating gradients. However, the training and validation error are almost identical throughout training. This is expected, as the validation and training trials only differ in the portion of the local field potential that was used as reference signal for the network, and in the seed used for generating the randomised stimulus onsets and offsets.
(TIF)

**S2 Fig. Dynamics of unconstrained networks are similar to low-rank RNNs.** A) We also trained RNNs without rank constraint. For these networks initial entries in the recurrent weight matrix $J$ were drawn from a zero-mean Gaussian with variance $\frac{g^2}{N}$. Here, we add a regularisation term to the loss, which keeps the average firing rates close to 0, to avoid a rate-coding solution (Eq 6). We set $g$ to 0.6 and took a learning rate of 0.001, with the training setup otherwise as for the low-rank networks (Training). In order to find a basis similar to the one used for plotting the dynamics of the low-rank RNN we took the following approach. First we calculated basis vectors for the activity due to recurrent dynamics: $\mathbf{J}\tanh(\mathbf{x}(t))$, by performing a Principal Component Analysis (singular vector decomposition): $\mathbf{U\Sigma V^T} = \mathbf{J}\tanh(\mathbf{X})$, where $\mathbf{X}$ is an $N \times 2\mathcal{T}$ matrix containing the activity of all units for one period of oscillation of each type of trial (stimulus $a$ and $b$). We took the first two columns (principal components), $\mathbf{u}_1$ and $\mathbf{u}_2$, of $\mathbf{U}$, as well as the input vector $\mathbf{I}^{(osc)}$ orthogonalised with respect to these two principal components as basis for $\mathbf{X}$. This basis retains 77% of the variance of $\mathbf{X}$ (measured by $r^2$). Since now, similar to the low-rank case, we can write the projection of $\mathbf{x}(t)$ on $\mathbf{I}^{(osc)}_\perp$ as a function of $\theta$, we can plot trajectories with coordinates $(\theta, \mathbf{u}_1^T\mathbf{x}, \mathbf{u}_2^T\mathbf{x})$, and we obtain two stable cycles, linked in phase-space, as in Fig 2. B) Here we train full-rank networks without regularisation, and find dynamics lying on two non-linked cycles, similar to the rate-coding model in Fig 2. The basis is constructed similar to A), here explaining 87% of the variance.
(TIF)

**S3 Fig. Influence of the training setup on the network solution.** A) Here we analysed to what degree a model will learn a phase- versus a rate-coding solution, as a function of the training setup and initialisation. To quantify to what degree either solution is learned, we first computed for all units in a model the absolute difference between their (normalized) rate after stimulus $a$ and their rate after stimuli $b$, as well as the absolute phase difference between trials with either stimulus. To get one measure of rate coding per model, we then calculated the mean over the absolute rate differences of all units, and similarly took the mean absolute phase differences as a measure of phase coding. We plot here in the first row these two measures for 6 models for each combination of the following conditions: ranks 1,2,3, full rank (columns); output during training is $\mathbf{Wx}$ (*lin-out*) or $\mathbf{W}\tanh(\mathbf{x})$ (*tanh-out*); regularisation that penalises deviations from the mean (Eq 6) is used (*reg*) or not (*no-reg*). All of these models use the default initialisation used in the manuscript and were trained for 100 epochs. Additionally we plot the validation loss after training in the second row. We observe that for rank 1 only models with *tanh-out* learn the task, and that these models learn a purely rate-coding solution. This is in line with our theory, as we previously showed that the phase-coding models couples its autonomously generated oscillations. Autonomous oscillations can only be generated by models of rank 2 and higher. For rank 2 and higher we can see that regularised models with a linear readout during training will generally learn a phase-coding solution, whereas models with a non-linear readout and no regularisation will generally learn a rate-coding solution. B) We repeated the experiment, but now initialised each network to start out with oscillatory dynamics. We can do this by initialising the weight matrix with a pair of complex conjugate eigenvalues with real part larger than 1 [30, 42] (this is only possible for rank 2 and higher). We can observe that models with oscillatory initialisation are generally more likely to learn a phase-coding solution; in particular all rank 2 models learn a purely phase-coding solution.
(TIF)

**S4 Fig. Rate-coding solution in detail.** A) We here detail a simple rate-coding model that performs the working memory task. Such a model consists of two populations, one responsible for an oscillatory output in-phase with the reference oscillation, and one responsible for the

anti-phase output oscillation. Depending on which stimulus was last seen, either population is dominating the readout. B) We reverse engineered a trained RNN with rank-1 connectivity. Due to the rank constraint, in the absence of any input, the trained model implements a one-dimensional dynamical system (with variable $\kappa$). We can plot the change in $\kappa$ as a function of $\kappa$ and observe the model learned to have three stable fixed points (left). We can fit a mixture of two Gaussians to the connectivity (**I** = input, **n**, **m** = left, right connectivity vector, **w** = output) which indicates that the model includes one population responsible for the fixed point at 0 (negative coupling between the connectivity vectors **m**, and **n**), while contributing little to the dynamics at the other fixed points (Due the large variance of **m**, the units of this population will have saturated rates when $\kappa$ is away from 0). This population produces an in-phase oscillation, due to the positive covariance between input ($\mathbf{I}^{(osc)}$) and output weights (**w**). The other population is responsible for the two stable fixed points away from 0 (positive coupling between the connectivity vectors **m**, and **n**, small variance of **m**), and produces the anti-phase oscillation. Note that this model is almost identical to the one described in [30] Fig 4 (but with oscillations added on top). C) We can make the description from the previous panel more formal by engineering a rate-coding model. We initialised two covariance matrices (right), creating a two population model with similar dynamics as in the previous panel (left). D) We plot the latent dynamics (top row) and output (bottom row) given by the mean-field equation describing the network activity in the limit of infinite units (blue, red), as well as finite size simulations (grey). For the finite size simulations, we used models (N = 4096) with weights sampled from the mixture of Gaussians. We can see that in the absence of input, depending on the initial state the model will go to either the fixed point at 0, corresponding to an in-phase oscillation, or one of the two fixed points away from 0, corresponding to an anti-phase oscillation, respectively. For the second and third column, we provided the network with stimulus *a*, and *b* respectively, for the first .15s of the trial. Stimuli can successfully steer the network to the right fixed point and output oscillation phase, irrespective of initial condition.
(TIF)

**S5 Fig. Converged RNN dynamics match those of coupled oscillators.** We simulated two coupled oscillators with the coupling function extracted from a trained network (Fig 3B). The two oscillators represent the RNN's phase ($\phi$) and the reference oscillation phase ($\theta$). Starting simulations from various initial conditions demonstrates that the coupling function induces bistability, as all simulations converge to one of two stable cycles on the torus. Furthermore, the convergent trajectories of the coupled oscillators are a close match to those of the full RNN projected into the same space ($\phi$, $\theta$), indicating that the coupled oscillator description is appropriate for the RNN.
(TIF)

**S6 Fig. Connectivity of trained models.** A) In order to study the connectivity of trained models, we fitted a mixture of Gaussians with 1 to 7 mixture components to the connectivity vectors of three different models [26], each trained with LFP data from a seperate rat. For this, we used variational inference with a Gaussian prior on the mean with precision $10^6$ and mean 0. After each fit, we resampled the weights 30 times and computed the loss over a batch of 128 trials with a pure sine wave as reference oscillation. Although for no amount of components we reliable were able to resample functioning models, from 3 components onwards a small fraction of sampled model have a comparable loss to the original trained model (red line). B) The covariance structure when fitting three components to weights of a trained network give us some hints as to what is needed to get a functioning model (top: covariances, bottom: pair plots). One component is unconnected (zero covariance) to the reference oscillation and has a skew-symmetric structure between the singular vectors. This structure generates oscillations

[30, 42, 55]. The other two components, each connected to one stimulus and with opposite covariance between the input and singular vectors, together implement the coupling function. (TIF)

**S7 Fig. Connectivity of mean-field model.** The designed covariance structure for the reduced model shown in Fig 4D and 4E, which leads to dynamics similar to the trained models. It also shares connectivity structure with trained models (S6B Fig), namely having one component unconnected to the reference oscillation that autonomously generates oscillations, and having the other two components, each connected to one stimulus, implement the coupling function. (TIF)

**S8 Fig. Geometry is preserved in a model coding for 4 stimuli, but more coupling populations are required.** A) Our model can be straightforwardly extended to code for more than 2 stimuli. Here, we trained a network to maintain 1 of 4 stimuli at a given trial, by producing an output oscillation at $-0.2\pi$, $-0.7\pi$, $-1.4\pi$ or $-1.7\pi$ radians offset with respect to the reference LFP, for stimulus $a$, $b$, $c$ or $d$, respectively. Again we find a stable limit cycle for each stimuli, which form linked cycles in phase-space. B) We now reverse engineered this model further, in particular we also found a reduced description of a model coding for 4 stimuli, in terms of 5 subpopulations. As before, 1 population generates oscillations, but we now have 4 coupling populations. Each stimulus inhibits all but one coupling population, so that the remaining coupling population guides the network dynamics to the limit cycle corresponding to the presented stimulus. C) The reduced model is specified by a mixture of 5 Gaussians, here we show the covariances that can be used to generate the connectivity of a model coding for 4 stimuli. D) Besides the connectivity with respect to the input, the recurrent dynamics (given by the overlaps, or covariances, between the connectivity vectors **m**'s and **n**'s) also had to be adjusted in order to allow memorising 4 stimuli instead of 2. We showed in our manuscript that rank 2 phase-coding models can described well by their coupling function, which gives the dynamics of the model as a function of the phase of the input oscillation ($\theta$) and the phase of its internal oscillation ($\phi$). The two limit cycles in the model described in the main text originated from the $\sin(2\phi)$ and $\cos(2\phi)$ terms in the coupling function. To code for 4 stimuli, we need higher frequency terms (e.g. $\cos(4\phi)$) in the coupling function. To understand how these originate from the recurrent connectivity, we decompose the coupling function. The coupling function generated by multiple populations is simply a summation over the coupling functions generated by the individual subpopulations. These in turn can be decomposed as the product of a linear part stemming from the covariances between the connectivity **m**'s and **n**'s of the subpopulations, and a non-linear part (the gain) that relates to how saturated neurons in this populations rates are (a neuron is saturated when its activity is at the flat part of the tanh non-linearity). Here we show that two populations that saturate at opposite phases of $\phi$ are enough to create a coupling function containing higher frequency terms, leading to dynamics with four limit cycles. (TIF)

**S9 Fig. Phase precession through translation of the coupling function.** A) Phase precession in rat hippocampus entails a place cell changing its relative phase of firing as a result of rat moving through a place field [10]. We here show how to create a network that changes its relative phase of oscillation depending on a continuous valued stimulus input. We setup a network with connectivity drawn from a mixture of three Gaussians, again with two components implementing a coupling function and one component implementing an oscillator. The network receives a sinusoidal reference input with phase $\theta$ as $\sin(\theta)$, $\cos(\theta)$ through input vectors $\mathbf{I}^{(osc_a)}$, $\mathbf{I}^{(osc_b)}$ respectively, as well as continuous valued stimulus input, representing place field

position, $s(t) \in [0, 1)$ as $\sin\left(s(t)\frac{1}{2}\pi\right)$, $\cos\left(s(t)\frac{1}{2}\pi\right)$ through input vectors $\mathbf{I}^{(s_a)}$, $\mathbf{I}^{(s_b)}$ respectively. The oscillator component has connectivity equal to S7 Fig, whereas for the coupling components $(p_2, p_3)$ we define the covariance matrices as follows: For the off-diagonal elements, $\sigma^{(p_2)}_{n^{(1)}I^{(osc_b)}} = -\sigma^{(p_2)}_{n^{(2)}I^{(osc_a)}} = \sigma^{(p_3)}_{n^{(2)}I^{(osc_a)}} = \sigma^{(p_3)}_{n^{(1)}I^{(osc_b)}}$, otherwise 0. For the diagonal elements, component two has zero variance for (is unconnected to) $\mathbf{I}^{(s_a)}$ and component three has zero variance for $\mathbf{I}^{(s_b)}$. This gives a coupling function of the form:

$$g(\theta, \phi) = \sin(\theta - \phi)\frac{a}{\sqrt{b + c\sin\left(\frac{\pi}{2}s(t)\right)^2}} + \cos(\theta - \phi)\frac{a}{\sqrt{b + c\cos\left(\frac{\pi}{2}s(t)\right)^2}},$$

for constants $a$, $b$, $c$ that depend on the exact values of the variances and covariances of the mixture components. For large $c$ and small $b$, the coupling function will change from $a\sin(\theta - \phi)$ to $a\cos(\theta - \phi)$ as $s(t)$ changes from 0 to 1. We plotted here, the coupling function of a network (N = 2056 units) with connectivity drawn from mixture of Gaussians as described above, for $s(t) = 0$ and $s(t) = 1$. The coupling function translates between these two states as a consequence of position input $s(t)$. B) Different trials on which $s(t)$ is tonically presented at different values lead the the network locking to a unique phase difference with respect to the reference oscillation. In phase-space, this appears as the stable cycle shifting along the surface of a torus. (TIF)

## Acknowledgments

We thank Richard Gao, Kabir Dabholkar and Stefanie Liebe for insightful discussions, Pedro J. Gonçalves, Elia Turner and Zinovia Stefanidi for feedback on the manuscript, and the International Max Planck Research School for Intelligent Systems (IMPRS-IS) for supporting MP.

## Author Contributions

**Conceptualization:** Matthijs Pals, Jakob H. Macke, Omri Barak.

**Formal analysis:** Matthijs Pals.

**Funding acquisition:** Jakob H. Macke, Omri Barak.

**Software:** Matthijs Pals.

**Supervision:** Jakob H. Macke, Omri Barak.

**Writing – original draft:** Matthijs Pals, Omri Barak.

**Writing – review & editing:** Jakob H. Macke.

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
