## [Decision Letter · Decision Letter 0]

19 Nov 2023

Dear Pals,

Thank you very much for submitting your manuscript "Trained recurrent neural networks develop phase-locked limit cycles in a working memory task" for consideration at PLOS Computational Biology. As with all papers reviewed by the journal, your manuscript was reviewed by members of the editorial board and by several independent reviewers. The reviewers appreciated the attention to an important topic. Based on the reviews, we are likely to accept this manuscript for publication, providing that you modify the manuscript according to the review recommendations.

Sincerely,

Lyle Graham

Section Editor

PLOS Computational Biology

Reviewer's Responses to Questions

**Comments to the Authors:**

Reviewer #1: This manuscript investigates the dynamics of recurrent neural networks (RNNs) trained on a working memory-like task, in which an RNN is trained to produce an oscillatory signal that goes in phase or out of phase with a reference oscillation, depending on a (transient) stimulus input (with one stimulus eliciting an in phase, and the other an out-of-phase oscillation). The authors find two (dynamically distinct) solutions by which the RNN can solve this task and go into an in-depth analysis on the solution that uses phase-coding. They then dissect the dynamics in detail and reverse engineer a simplified solution to the working mechanisms of the RNN, a coupled oscillator. This could be a simplified mechanism of how the brain solves certain tasks related to phase-coding.

Overall, the paper is well written, the figures are excellent, and I liked how the authors constrained the RNNs to make them expressive, yet interpretable in low (visualizable) dimensions. I must say though that I am no expert on the precise methods used to investigate stability, cycles, and low dimensional manifolds, although I think I understood the gist. I would recommend the paper for publication, just have several minor questions and remarks.

1. The authors state there were two solutions found to solve the task – I wonder which one was more frequent? Most importantly, I wonder how the solutions were more generally affected by the applied connectivity constraints? Is there any lack in expressive power of the RNN through this approach? Or will certain solutions be favored over others?

2. Just to be sure I understood the rationale: the underlying (neurobiological) assumption of the way the task is composed, is that some part of the brain or neuronal subpopulation acts as an oscillatory reference signal (e.g., CA1), and that another population is coupled to this reference signal as well as external inputs, and by this can generate a second oscillatory signal (e.g., in-phase/out-of-phase, different limit cycles) used for phase-coding. The idea of the task is that the RNN is trained to produce this secondary signal? Or, in other words, what is the rationale behind making the RNN reproduce periodic behavior in response to the stimulus? Would appreciate to make this a bit clearer in the text, perhaps with a cognitive example.

3. Can you quantify this statement: “Networks were able to successfully learn the task” (line 55). I guess in this application it is not necessary that all networks perform well, but the networks that do, should have a reasonably high “hit rate”?

4. Line 261: “10 % served as validation set”, so were the final analyzed models selected based on minimizing the test error?

Other remarks

5. Line 36: should be “analyses”

6. Line 249: missing brackets

7. Line 247: a word too many?

8. Line 264: one “we” too many

9. Line 267: one “by” too many

10. Taylor expansion equation, misses a period after “h.o.t”, and punctuation missing in the sentence after

11. Line 275: “Euler” method

Reviewer #2: This paper presents a new hypothesis, gleaned from RNNs trained on a working memory task, about how the brain might encode stimuli using phase-coding alongside neural oscillations. I think this paper's results will be of broad interest to a computational audience, and the authors did a wonderful job of presenting their findings at multiple levels (both technical and high-level). My main suggestions are that the authors more clearly explain 1) why/when their networks arise at two solutions, and 2) why only one of those solutions is of interest. I think these two additions would make the paper's findings much more comprehensive/complete. Otherwise I have only minor comments/suggestions.

major comments:

line 69: what does "depending on the training setup" mean? are these different solutions just due to the network's random initialization? (coming back to this later:) Fig S3 seems to explain it. am i correct in understanding that the network's solution is totally determined by whether there is a nonlinearity in the objective? as the results are presently presented in the main text, it's not clear that the authors understand why the two solutions occur, so I think it would strengthen their presentation to go into this more in the main text.

the authors mention that the second solution resembles "previous work on fixed point dynamics in RNNs." i didn't really follow what this means. in general, this felt like a pretty brief dismissal of one of the two identified network solutions.

minor comments:

could the authors add to the results text a brief rationale for why the network is provided with an oscillatory reference input (e.g., near equation 1)?

line 50: here the paper introduces the use of LFPs. could the authors provide some context for why this is the right thing to include in the network? also, does the inclusion of LFPs in u(t) mean we should think of x(t) as also being an LFP signal? or should x(t) be thought of as firing rates? (i guess my general point is that it could be helpful to have some high-level/systems-level explanation of how the authors are imagining LFPs and firing rates to interact.)

Fig 2A and Fig 2D took me a long time to parse. It was confusing to me that the caption for Fig 2A referred to the "top row" but not the "bottom row", because I assumed "top row" meant the top row _of Fig 2A_, meaning the top two units. it would also help to add a text title to Fig 2A that says "Solution 1" and another one to Fig 2D that says "Solution 2." And then label each subpanel as Unit 1, Unit 2, etc.

after seeing Fig 2 (which I found really clear and interesting, once I understood what the subpanels meant), i found myself wondering how I can interpret these two solutions as valid solutions given the network's objective. currently, the training objective is only presented in methods. it could be helpful to have some explanation of the objective in results, and then help the reader understand why the two network solutions "make sense" given this objective. for example, is it possible to explain why the unit activity in Fig 2D is a reasonable solution to this task?

the explanation of poincare maps and the cross-section Q went over my head. the function (?) P in the equation after line 85 is also not defined. also, the vector K_c is in S, but I don't know what S is, nor does I know what an intersection is. I'm not sure if me understanding all of these things is essential for me to get the point, so you could potentially move this (along with more details) to methods. but either way a little more hand-holding would help a lot.

Fig 3 as a whole is an excellent graphical depiction of what is going on, and the text for this part is, on the whole, very clear. loved it!

do the authors know what the importance of the network having rank 2 dynamics is? i am curious why the authors chose that particular rank, and also if this "matters" in the sense of the network solutions. e.g., does solution 1 vs 2 dominate if the dynamics had rank 1, or rank 3, etc?

the model shown in Fig 4D had me wondering how easily this would scale to multiple stimuli. e.g., from Fig 4D it seems like you'd need a new coupling population (or maybe even more) for a third stimulus--is this true? i see that Fig S7 shows this is possible in principle. but could the authors explain how their model needs to change to accommodate more stimuli, e.g., does the model in Fig 4D need to be changed?

**Have the authors made all data and (if applicable) computational code underlying the findings in their manuscript fully available?**

Reviewer #1: Yes

Reviewer #2: Yes

PLOS authors have the option to publish the peer review history of their article (what does this mean?). If published, this will include your full peer review and any attached files.

Reviewer #1: No

Reviewer #2: No

Figure Files:

Data Requirements:

Reproducibility:

References:

---

## [Decision Letter · Decision Letter 1]

22 Jan 2024

Dear Pals,

We are pleased to inform you that your manuscript 'Trained recurrent neural networks develop phase-locked limit cycles in a working memory task' has been provisionally accepted for publication in PLOS Computational Biology.

Best regards,

Lyle Graham

Section Editor

PLOS Computational Biology

Reviewer's Responses to Questions

**Comments to the Authors:**

Reviewer #1: Thank you for addressing my questions.

Reviewer #2: Thank you for your replies to my comments and the updates you made to the manuscript. I found the updated manuscript to be much improved.

**Have the authors made all data and (if applicable) computational code underlying the findings in their manuscript fully available?**

Reviewer #1: Yes

Reviewer #2: Yes

PLOS authors have the option to publish the peer review history of their article (what does this mean?). If published, this will include your full peer review and any attached files.

Reviewer #1: No

Reviewer #2: No

---

## [Editor Report · Acceptance letter]

31 Jan 2024

PCOMPBIOL-D-23-00585R1 

Trained recurrent neural networks develop phase-locked limit cycles in a working memory task

Dear Dr Pals,

I am pleased to inform you that your manuscript has been formally accepted for publication in PLOS Computational Biology. Your manuscript is now with our production department and you will be notified of the publication date in due course.

With kind regards,

Zsofi Zombor
